# Increasing Incidence and Shifting Epidemiology of Candidemia in Greece: Results from the First Nationwide 10-Year Survey

**DOI:** 10.3390/jof8020116

**Published:** 2022-01-26

**Authors:** Vasiliki Mamali, Maria Siopi, Stefanos Charpantidis, George Samonis, Athanasios Tsakris, Georgia Vrioni

**Affiliations:** 1Department of Microbiology, Tzaneio General Hospital, 18536 Piraeus, Greece; mamalivasiliki@outlook.com.gr; 2Clinical Microbiology Laboratory, “Attikon” University General Hospital, Medical School, National and Kapodistrian University of Athens, 12462 Athens, Greece; marizasiopi@hotmail.com; 3Department of Microbiology, “Elena Venizelou” Maternity Hospital, 11521 Athens, Greece; steve_kavala2@hotmail.com; 4Department of Internal Medicine, School of Medicine, University of Crete, 71003 Heraklion, Greece; samonis@med.uoa.gr; 5Department of Microbiology, Medical School, National and Kapodistrian University of Athens, 11527 Athens, Greece; atsakris@med.uoa.gr

**Keywords:** candidemia, epidemiology, Greece, species distribution, antifungal resistance

## Abstract

Globally, candidemia displays geographical variety in terms of epidemiology and incidence. In that respect, a nationwide Greek study was conducted, reporting the epidemiology of *Candida* bloodstream infections and susceptibility of isolates to antifungal agents providing evidence for empirical treatment. All microbiologically confirmed candidemia cases in patients hospitalized in 28 Greek centres during the period 2009–2018 were recorded. The study evaluated the incidence of infection/100,000 inhabitants, species distribution, and antifungal susceptibilities of isolated strains. Overall, 6057 candidemic episodes occurred during the study period, with 3% of them being mixed candidemias. The average annual incidence was 5.56/100,000 inhabitants, with significant increase over the years (*p* = 0.0002). *C. parapsilosis* species complex (SC) was the predominant causative agent (41%), followed by *C. albicans* (37%), *C. glabrata* SC (10%), *C. tropicalis* (7%), *C. krusei* (1%), and other rare *Candida* spp. (4%). *C. albicans* rates decreased from 2009 to 2018 (48% to 31%) in parallel with a doubling incidence of *C. parapsilosis* SC rates (28% to 49%, *p* < 0.0001). Resistance to amphotericin B and flucytosine was not observed. Resistance to fluconazole was detected in 20% of *C. parapsilosis* SC isolates, with a 4% of them being pan-azole-resistant. A considerable rising rate of resistance to this agent was observed over the study period (*p* < 0.0001). Echinocandin resistance was found in 3% of *C. glabrata* SC isolates, with 70% of them being pan-echinocandin-resistant. Resistance rate to this agent was stable over the study period. This is the first multicentre nationwide study demonstrating an increasing incidence of candidemia in Greece with a species shift toward *C. parapsilosis* SC. Although the overall antifungal resistance rates remain relatively low, fluconazole-resistant *C. parapsilosis* SC raises concern.

## 1. Introduction

Candidemia is among the leading nosocomial bloodstream infections (BSIs) globally, representing the most frequently encountered manifestation of invasive candidiasis. Regardless of the ongoing advances in treatment algorithms and availability of new antifungal agents with improved spectrum and potency, it remains associated with high mortality rates [1,2]. Meanwhile, *Candida* BSIs contribute towards prolonged hospital stay, imposing a considerable economic burden on the health care systems, since $157,574/patient are spent on their management, excluding attributable costs ($82,320/patient) [3]. Of note, prompt management of candidemia is crucial to improve clinical outcome, given that delays in time to treatment initiation have been associated with increased mortality [4,5]. On these grounds, empirical antifungal therapy with echinocandins, fluconazole, or a lipid formulation of amphotericin B is often prescribed as soon as possible to patients with traditional risk factors for developing the infection, long before definitive identification and susceptibility data become available [6]. Therefore, in-depth understanding and monitoring of temporal local epidemiological and in vitro antifungal susceptibility trends is imperative in terms of guiding informed therapeutic decisions.

The epidemiology of candidemia may change over time, whilst its patterns can present significant geographical, centre-to-centre, and even unit-to-unit variability, due to local factors and practices [2]. Although *C. albicans* is still considered the main causative pathogen, a progressive shift to non-*albicans Candida* (NAC) spp. is currently recorded in most parts of the world [2,7]. At the same time, emerging azole and echinocandin resistance among commonly isolated species displays major challenges for therapeutic strategies [2,7], whereas novel pathogenic species with multi-resistance profiles, such as *C. auris*, set a worrisome trend and amplify the call for alertness [8]. Taking into account that each *Candida* spp. has a unique virulence potential, clinical characteristics, and antifungal susceptibility profile, the evolving epidemiology of *Candida* BSIs may have different implications regarding their management, thus reinforcing the need for comprehensive regional and local epidemiological surveillance with provision of feedback at regular intervals.

To date, although a plethora of studies have addressed the menace of candidemia worldwide, data on its contemporary prevalence and resistance patterns in Greece are relatively few, while published reports are limited to single-centre studies, small geographical regions, and distinct patient populations [9]. As a consequence, results may not be generalizable regarding all candidemic patients; neither can conclusions be extrapolated regarding other centres as the epidemiology of the infection can be highly institution-specific. Based on these grounds, a nationwide study was conducted to describe the current epidemiological characteristics of *Candida* BSIs in Greece and the susceptibilities of the causative strains to antifungal agents, and to provide up-to-date evidence for empirical treatment.

## 2. Materials and Methods

Study design. A retrospective, laboratory-based study of patients with candidemia was conducted in 28 Greek hospitals during the period 2009–2018. Namely, in the study were included 26 public hospitals (16 general, 6 university-affiliated, 2 cancer, 1 maternity, and 1 military) and two private, regional, general hospitals. The participating centres were geographically distributed in densely populated metropolitan areas across the country covering all seven Greek regional administrative health authorities [10]: 20 of them located in the Attica region (35% of the country’s population), 5 in Northern Greece (Alexandroupolis, Ioannina, Serres and Thessaloniki), 1 in Central Greece (Larissa), 1 in Southern Greece (Patras), and 1 on the island of Crete (Heraklion) (Figure 1). The study was planned and registered as “HSoMM (Hellenic Society of Medical Mycology) Candi-Candi Network: an observational study”.

Candidemia was defined as the recovery of *Candida* spp. from at least one blood culture set during hospitalization. A case was defined as intensive care unit (ICU)-acquired candidemia if it occurred >48 h after ICU admission [11]. Subsequent positive blood cultures with the same *Candida* spp. from a single patient were considered as a new episode if the episodes occurred >4 weeks apart, along with the clearance of the prior blood culture and resolution of all clinical features of the infection. Blood cultures yielding different *Candida* spp., independently of the time interval between the new and the prior positive blood culture, were considered to represent new episodes. Patients with mixed candidemia (MC), identified as the isolation of two different *Candida* spp. from a single blood culture sample, were included. Medical unit at the onset of infection and mycological findings (species identification and in vitro susceptibility profile of the causative pathogens) were retrospectively obtained from individual laboratory records or the microbiological laboratory computerized database of each participating hospital.

Identification and antifungal susceptibility testing. Fungal isolates were identified to species level as per hospital protocol by germ tube production, colony colour, and morphology on chromogenic agar (4/28 participating centres) and analysis of biochemical pattern using automated systems (Vitek^®^2 (BioMeriéux, Marcy l’Etoile, France), BD Phoenix^TM^ (BD, Sparks, MD, USA)) or commercially available kits (Auxacolor (Bio-Rad, Hercules, CA, USA), MicroScan Rapid Yeast Identification (Beckman Coulter, Brea, CA, USA), API 20C AUX and ID 32C (BioMeriéux, Marcy l’Etoile, France)). 

In vitro susceptibility was determined as per hospital protocol using the automated susceptibility testing system Vitek^®^2 (BioMeriéux, Marcy l’Etoile, France), gradient concentration strips (Etest (BioMeriéux, Marcy l’Etoile, France), MTS (Liofilchem, Roseto degli Abruzzi, Italy)), the colorimetric assay Sensititre YeastOne (SYO; Thermo Fisher Scientific, Cleveland, OH, USA), or the broth microdilution reference method following the European Committee on Antimicrobial Susceptibility Testing (EUCAST) guidelines [12,13]. Vitek^®^2 [14,15,16], Etest/MTS [17], and SYO [16,18] have been shown to give comparable results to those obtained by the Clinical and Laboratory Standards Institute (CLSI) procedure for antifungal susceptibility testing of yeasts. Thus, for the categorization of the isolates, the M60 CLSI species-specific clinical breakpoints were applied [19] and in the absence of those method-specific (CLSI (for data obtained by Vitek^®^2) [20], Etest [21,22,23], or SYO [22,24,25]), epidemiological cut-off values (ECVs) were used to differentiate wild type (WT) and non-WT phenotypes. 

Statistical analysis. The incidence of candidemia was expressed as the ratio of *Candida* BSI episodes per 100,000 inhabitants, based on the annual population data extracted from the World Bank subnational population database [26], whereas its trends over time were evaluated by linear regression analysis and ANOVA, followed by post-test for linear trend. Medians and interquartile ranges (IQR) were calculated for continuous variables, while numbers and percentages were calculated for categorical parameters. Categorical variables were compared by Pearson’s chi-square test. In any case, a two-tailed *p* value of <0.05 was considered to reveal a statistically significant difference. All data were analysed using the statistics software package GraphPad Prism, version 8.0, for Windows (GraphPad Software, San Diego, CA, USA).

## 3. Results

Candidemia incidence. Over the 10-year study period, a total of 6057 *Candida* BSIs were recorded, whereas no patient had separate episodes with distinct *Candida* spp. The median (range, IQR) number of cases reported per year was 639 (373–811, 214). The distribution of episodes in age categories was 132 (2%) in neonates, 69 (1%) in children, and 5856 (97%) in adults. The overall incidence of candidemia was 5.56/100,000 inhabitants with a significant increase over the years, particularly between 2009–2011, 2012–2014, and 2015–2018 (3.75, 5.83, and 7.01/100,000 inhabitants, respectively; *p* = 0.0002). The highest number of cases was determined in 2017, with a ratio of 7.54/100,000 inhabitants, while the lower was in 2009 with 3.36/100,000 inhabitants (Figure 2). Almost half (2665/6057; 44%) of the episodes occurred in patients admitted in internal medicine wards (IMWs), 1999/6057 (33%) in ICUs (1827/1999; 91% in adult, 40/1999; 2% in paediatric, 132/1999; 7% in neonatal), and 1393/6057 (23%) in surgery wards (SWs). The median (range, IQR) number of candidemias reported per year was 284 (219–306, 87), 209 (169–227, 58), and 146 (111–160, 49) in IMWs, ICUs, and SWs, respectively.

*Candida* species distribution. A total of 6239 non-duplicate *Candida* spp. isolates were recorded, 37% (2333/6239) were *C. albicans* and 63% (3906/6239) were NAC spp. (*p* < 0.0001). Of note, *C. albicans* rates decreased from 2009 to 2018 (48% to 31%) in parallel with a doubling of *C. parapsilosis* species complex (SC) rates (28% to 49%; *p* < 0.0001) (Figure 2). No significant change was observed in the proportions of other *Candida* spp. Overall, *C. parapsilosis* SC accounted for the majority of isolates (*n* = 2582; 41%) followed by *C. albicans* (*n* = 2333; 37%), comprising 78% of all *Candida* BSIs. Other commonly encountered *Candida* spp. included *C. glabrata* SC (*n* = 594; 10%), *C. tropicalis* (*n* = 423; 7%), and *C. krusei* (*n* = 86; 1%). Other species, such as *C. lusitaniae* (*n* = 73), *C. famata* (*n* = 36), *C. guilliermondii* SC (*n* = 28), *C. dubliniensis* (*n* = 22), *C. sake* (*n* = 16), *C. kefyr* (*n* = 12), *C. lipolytica* (*n* = 8), *C. pelliculosa* and *C. rugosa* (each *n* = 6), *C. ciferrii* and *C. zeylanoides* (each *n* = 3), *C. norvegensis* and *C. sphaerica* (each *n* = 2), *C. globosa*, *C. intermedia*, *C. pulcherrima,* and *C. utilis* (each *n* = 1), were relatively rare (*n* = 221; 4%). Depending on the year, the percentage of isolated species varied from 31% to 48% for *C. albicans*, 28% to 49% for *C. parapsilosis* SC, 8% to 12% for *C. glabrata* SC, 5% to 11% for *C. tropicalis*, 1% to 3% for *C. krusei,* and 3% to 8% for other *Candida* spp. (Figure 2).

Temporal distributions of *Candida* spp. in different hospital wards are shown in Figure 3. The proportion of *C. albicans* versus NAC spp. isolates differed significantly in all admission wards at the time of diagnosis (IMWs 39% versus 61%, SWs 38% versus 62% and ICUs 32% versus 68%, respectively; *p* < 0.0001). A higher proportional increase of *C. parapsilosis* SC isolation rate was recorded in wards, from 25% and 18% in 2009 to 44% and 50% in 2018 in IMWs (*p* < 0.0001) and SWs (*p* = 0.0003), respectively, than in the ICUs, from 42% in 2009 to 59% in 2018 (*p* = 0.002). Distinct unit-related patterns of species distribution were not observed (*p* = 0.96). *C. parapsilosis* SC was the most frequently seen species in adult ICUs (48%) and almost equally distributed with *C. albicans* in IMWs (37% versus 39%, respectively) and SWs (39% versus 38%, respectively), while almost half (97/201; 48%) of the neonatal/paediatric patients were infected with *C. parapsilosis* SC isolates.

Out of 4/28 participating hospitals using chromogenic agar for *Candida* as an additional primary isolation medium, candidemia with two distinct *Candida* spp. was determined in 24/795 (3%) cases, with *C. albicans* being the species most frequently isolated in combination with others (17/24; 71%). In particular, 13 patients (54%) presented with *C. albicans* and *C. parapsilosis* SC, 3 (13%) with *C. glabrata* SC and *C. parapsilosis* SC, 2 (9%) with *C. albicans* and *C. glabrata* SC, 1 (4%) with *C. parapsilosis* SC and *C. tropicalis*, 1 (4%) with *C. albicans* and *C. krusei*, 1 (4%) with *C. albicans* and *C. lusitaniae*, 1 (4%) with *C. albicans* and *C. kefyr*, 1 (4%) with *C. parapsilosis* SC and *C. kefyr,* and 1 (4%) with *C. parapsilosis* SC and *C. sake*. The majority of the MC episodes occurred in patients hospitalized in IMWs (17/24; 71%), while 5/24 (21%) occurred in SWs and 2/24 (8%) in the ICUs. 

Antifungal susceptibility profile. The results of in vitro susceptibilities to antifungal agents were available for a subset of *Candida* bloodstream isolates, depending on the method used for susceptibility testing. Namely, 3615/6239 (58%) strains were tested using Vitek^®^2, and thus amphotericin B (AMB), flucytosine (5FC), micafungin (MFG), caspofungin (CAS), voriconazole (VRC) and fluconazole (FLC) minimum inhibition concentrations (MICs) were available. MIC values of these antifungals as well as anidulafungin (AFG), posaconazole (POS) and itraconazole (ITC) were obtained for 914/6239 (15%) and 469/6239 (7%) isolates by Etest/MTS and SYO, respectively. AMB, echinocandins and azoles (except for ITC) EUCAST MICs were determined for 51/6239 (1%) isolates. For the rest 1190/6239 (19%) isolates, susceptibility data were not available. 

The in vitro susceptibility results for each *Candida* spp. to antifungals are summarized in Table 1. Overall, the majority of isolates were susceptible/WT to the drugs tested. No resistance to AMB and 5FC was found.

(i).Azoles. For ITC and POS, resistant/non-WT strains were observed among *C. albicans* (4% and 3%, respectively), *C. parapsilosis* SC (7% and 5%, respectively), and *C. glabrata* SC (7% and 12%, respectively). Interestingly, a significant proportion of the ITC resistant/non-WT isolates, all recovered from IMWs and SWs patients were pan-azole-resistant/non-WT (54%, 41%, and 47% of *C. albicans*, *C. parapsilosis* SC, and *C. glabrata* SC, respectively). VRC-resistant/non-WT phenotypes were identified among strains of *C. albicans* (3%), *C. parapsilosis* SC (1%), *C. glabrata* SC (6%), and *C. tropicalis* (1%), while 7% and 10% of *C. parapsilosis* SC and *C. tropicalis* isolates, respectively, displayed elevated VRC MICs (0.25–0.5 mg/L), categorizing them as intermediate. Worryingly, reduced susceptibility to FLC was mostly seen. In particular, 3% of *C. albicans* (18% pan-azole-resistant/non-WT), 20% of *C. parapsilosis* SC, 5% of *C. glabrata* SC (3% pan-azole-resistant/non-WT), and 6% of *C. tropicalis* isolates were FLC-resistant, whereas 2% of *C. albicans* as well as *C. parapsilosis* SC and 8% of *C. tropicalis* isolates were categorized as intermediate. The FLC-resistant *C. parapsilosis* SC isolates were found in all units (48% in ICUs, 34% in IMWs and 18% in SWs) of the participating hospitals, presenting the 32%, 21%, and 20% of *C. parapsilosis* SC isolates recovered from candidemic patients admitted to ICUs, IMWs, and SWs, respectively. Alarmingly, their isolation rate was steadily rising throughout the study period: from 1% during 2009–2011 to 14% between 2012–2014 and further to 27% during 2015–2018 (*p* < 0.0001) (Figure 4). Moreover, they have shown different susceptibility profiles to other azoles; 3% were pan-azole-resistant/non-WT isolates, whilst those with the highest MICs for FLC (≥32 mg/L) were also VRC-resistant (20%).(ii).Echinocandins. All three echinocandins exhibited very good activity against most *Candida* spp., including *C. parapsilosis* SC isolates (100% susceptibility). Non-susceptible to AFG and MFG *C. albicans*, *C. tropicalis,* and *C. krusei* isolates remained below 2%. Of note, echinocandin resistance was found in 3% of *C. glabrata* SC isolates, whereof 70% demonstrated elevated MIC values for all echinocandins (AFG, CAS, and MFG MIC 0.5–1, 0.5–2, and 0.5 mg/L, respectively) but not to azoles. These strains were isolated from ICU patients hospitalized in different and far from each other medical centres and were distributed equally through the years of the study period (0–3 isolates annually; *p* = 0.80) (Figure 4). 

No significant trend over time was found in the annual susceptibility rates of *Candida* spp. to the rest of the antifungals.

## 4. Discussion

In light of the constantly evolving epidemiological landscape of candidemia worldwide with important implications in this infection’s management, continuous monitoring, specifically in previously under-investigated geographical areas, is warranted. Taking into account the existing literature, this is the first study aiming to determine the countrywide incidence of *Candida* BSIs in Greece, along with the sensitivity spectrum profiles of the etiological *Candida* spp. to antifungal agents. During the 10-year (2009–2018) period, a significant increase of the incidence of candidemia has been observed all over the country. During this period *C. parapsilosis* SC has emerged as a major causative agent of candidemia, now accounting for 41% of *Candida* bloodstream isolates. Worryingly, *C. parapsilosis* SC isolates have shown rising FLC resistance rate. 

A precise estimate of the global burden of candidemia is difficult to assess since long-term surveillance data are limited and highly heterogeneous [1,27]. In fact, reported incidence rates vary significantly between countries, ranging from 2.0 to 21.0/100,000 inhabitants [27], while differences may also occur within regions of the same country and between populations at risk [28,29]. Similar intracountry variations, attributed to differences in study periods, local practices, and antifungal drug use as well as distinct patient populations and degrees of illness severity have been recently described in a systematic review of the existing literature related to candidemia in Greece [9]. In a first attempt to depict the prevalence of serious fungal infections in this country, the incidence of *Candida* BSIs in ICU patients and in non-ICU immunosuppressed patients with haematological malignancies was estimated at 5.0/100,000 population (541 cases/year; 162 in ICU patients and 379 in non-ICU haematology patients) [30]. The present epidemiological survey of the Greek all cause patient population (all hospital units) is representative of the entire country over a 10-year period providing a more complete picture. The survey has revealed an average annual incidence of 5.56/100,000 inhabitants (639 cases/year; 284, 209, and 146 in IMWs, ICU, and SWs patients, respectively), which is in line with reports from Ireland (6.3) [31], Kuwait (5.29) [32], Mexico (5.0) [33], and Sweden (4.7) [34]. The incidence is lower than that reported in Thailand (13.3) [35], Hungary (11.0) [36], Denmark (8.13) [37], and Spain (8.1) [38], and higher than in Canada (2.91) [39], Australia (2.41) [40], Portugal (2.19) [41], and the Philippines (2.0) [42]. Of note, a recent epidemiological meta-analysis has shown that the overall pooled incidence rate of candidemia in Europe is 3.88/100,000 inhabitants per year, yet is significantly higher in the southern (5.29) than in the northern (3.77) or the western (2.5) European countries [1], corroborating the present finding. Several factors may contribute to the high degree of variability in population-based candidemia incidences, such as geographical and ecological parameters, lack of uniformity in monitoring and reporting systems, the overall health of the studied patient populations, demographic development and setting of local health care systems, implementation of infection control procedures, stewardship programs and educational campaigns, as well as differences in clinical management and antimicrobial prescription policies [1,2,43].

Of interest, the present findings have shown a significant increase in the incidence of candidemia. Namely the rates were 3.75, 5.83, and 7.01/100,000 inhabitants during the periods 2009 to 2011, 2012 to 2014, and 2015 to 2018, respectively (*p* = 0.0002). Notably, these findings are supported by other long-term, nationwide epidemiological surveillance studies of *Candida* BSIs in other European countries [44,45], highlighting the need for constant vigilance. A possible explanation for this rise is the growing number of patients at risk, taking into account that the European health care systems are called upon to provide services for an ever-increasing number of elderly and debilitated patients with complex and severe underlying disorders [46]. Prolonged hospital stay due to increased survival rates, expanding indications for treatment with immunosuppressive and antineoplastic drugs, increased number of solid organ and haematopoietic stem cell transplantation procedures, as well as extent use of indwelling medical devices, broad-spectrum antimicrobials, and parenteral nutrition, might also be considered as contributing factors [1]. Of note, Greece is particularly vulnerable to compound risks from discrepancies in the organization and resourcing of health care delivery practices, given that Greece, nowadays, like other Mediterranean countries, has medium-low-quality hospitals with a high occurrence of nosocomial infections [47]. Indeed, single-centre and regional studies from Italy have also reported similar increase in candidemia incidence over time [48,49]. This issue has experienced fluctuations since 2010, which may be explained by the unstable economic situation in Greece, due to the financial crisis which could have adversely influenced basic infection control measures and promoted the onset of hospital-acquired infections, like *Candida* BSIs as previously described [9]. In particular, during the study period, the number of Greek hospitals has been steadily decreased; in 2013 fell below 300, while in 2018 there were 271 hospitals all over the country, which is the lowest number in this study time interval. Correspondingly, the number of specialized hospital staff increased from 2000 to 2009, but it has been declining ever since [50]. At the same time, the number of patients hospitalized in public medical institutions was constantly increasing, resulting in an inadequate patient-to-nurse ratio, whereas there was deficiency of resources for medical care and training, as well as infection control programs accompanied by critical challenges in clinical management in regard to both the diagnosis (slow laboratory turnaround times, nonavailability of biomarkers) and treatment (specifically shortage of some antifungal agents) [51,52]. In fact, the increase in the incidence of candidemia reported in the present study coincided with the beginning (2012 to 2014) and the peak (2015 to 2018) of the financial crisis in Greece. Hence, it is apparent that severe socioeconomic events may influence the epidemiology of infectious diseases and lead to changes of the hospital settings that eventually will have deleterious impacts on human lives, bearing in mind that 29 patients die only in Europe from candidaemia every day [1]. 

Concurrently, a progressive rise of BSI attributed to NAC spp. has been recorded, consistent with the current local [9,53] and global epidemiological trends [1,2,7]. In particular, the overall ratio of NAC spp. versus *C. albicans* was 1.7 (*p* <  0.0001). The present data have illustrated that this disproportion was mainly due to the considerable prevalence of *C. parapsilosis* SC in all medical units during the last years, given that the frequency of *C. albicans* isolation decreased from 2009 to 2018 (48% to 31%) in parallel with a doubling of *C. parapsilosis* SC recovery (28% to 49%; *p* < 0.0001). The proportion of other NAC spp. such as *C. glabrata* SC and *C. tropicalis*, that increased in several North American [54] as well as Central/North European [37,44] and East/Southeast Asian countries [2,55], remained stable. Overall, *C. parapsilosis* SC was the predominant species (41%), followed by *C. albicans* (37%), *C. glabrata* SC (10%), *C. tropicalis* (7%), *C. krusei* (1%), and other rare *Candida* spp. (4%). Emerging species, such as *C. auris*, had not been detected. Therefore, during the study period, Greece had not yet been incorporated in the increasingly expanding map of countries where *C. auris* BSIs had been documented [8]. Taking into account the published literature, this is the first study describing the epidemiological shift in favour of *C. parapsilosis* SC as the main etiologic agent of candidemia in the Greek all cause patient population [9]. This shift has lately been evident in local scale. Namely, the most recent single-centre studies in all cause patient population of Athens (capital of Greece; 2009–2018) and Patras (southwestern Greece; 2009–2017) revealed that *C. parapsilosis* SC were almost equally distributed with *C. albicans*-driven BSIs (37% versus 41% and 37% versus 40%, respectively) [9,53]. Likewise, *C. parapsilosis* SC has outranked *C. albicans* in several geographical regions, such as South Africa [56] and South America (Brazil, Colombia, Peru) [57,58,59], or has comparable isolation rates [7] indicating that the incidence of this pathogen is continuously rising with consequent clinical importance.

The dominance of *C. parapsilosis* SC is worrisome since it has been associated with central venous catheter infections and the administration of parenteral nutrition due to its propensity to form tenacious biofilms, thus threatening ICU patients and newborns. In the present study, almost half (48%) of both the adult ICU and neonatal/paediatric patients were infected with *C. parapsilosis* SC isolates, aligning with other surveys [60,61]. Of note, a recent multicentre study on antifungal prescription in Greek hospitals showed that the use of echinocandins was significantly higher in ICUs as compared with those in all other departments [62]. It can be assumed that in some medical units FLC was used as prophylaxis and on that account critically ill patients hospitalized in ICUs were considered as eligible subjects to be treated with an echinocandin in case of suspected breakthrough fungal infection. In addition, the rising incidence of FLC-resistant *C. parapsilosis* SC isolates in Greek hospitals might explain the extensive use of echinocandins as pre-emptive or empirical treatment in the ICU setting [9,53]. Thus, the increasing selection pressure mediated by a larger use of echinocandins probably promoted this local epidemiological shift considering that the increased use of echinocandins has been correlated with an expansion of *C. parapsilosis* SC [9,48,63,64]. Furthermore, a higher proportional increase of *C. parapsilosis* SC isolation rate was recorded in wards. This rising burden may be related to changes in the hospital case mix with an expanding population of immunosuppressed and/or debilitated patients [46] in conjunction with moderate compliance with infection control prevention guidelines and basic measures, such as hand hygiene, in debt-stricken Greek hospitals [51,52]. Hence, in contrast to *C. albicans*-driven candidemia that is acquired mostly endogenously [65], the horizontal transmission potential of *C. parapsilosis* SC via contaminated medical equipment and hands of health care personnel, which may lead to crossover infections between patients and is often responsible for nosocomial cluster outbreaks, cannot be excluded. Intensity of the shift towards *C. parapsilosis* SC is of concern, given that it may provide a challenge for current antifungal treatment strategies and stresses the need for species identification and susceptibility testing as well as thorough consideration of the local epidemiology. 

Species-specific profiles may be used in an attempt to predict antifungal susceptibility, while awaiting susceptibility testing results [6]. Hence, accurate identification of the species implicated in candidemia is a crucial requisite for timely and optimally targeted antifungal treatment that in turn ensures better prognosis [4,5]. In recent years, matrix-assisted laser desorption-time-of-light mass spectrophotometry (MALDI-TOF MS) has emerged as a promising method for yeast identification. Nevertheless, this study evaluated 10-year retrospective epidemiological data, when MALDI-TOF MS was not broadly available [66]. Regarding the Greek hospital reality during the study period, identification of *Candida* isolates was made mainly with automated systems, while there was no capacity for performing molecular identification in laboratory routine [67]. Despite the fact that isolates belonging to different SCs can be reliably differentiated by assimilation methods, cryptic species within larger complexes exhibiting differences both in virulence and in the spectrum of antifungal resistance [68,69,70] may be missed. Although this can be considered as a limitation of the present study, it is partially offset by the low isolation rate of such species [68,69,70].

Since most of *Candida* spp. have undistinguishable colony morphologies on standard media that may not allow reliable differentiation, the routine use of chromogenic agars can increase significantly the detection rates of mixed species [71,72]. The incidence of MC may vary from 1.5% to 18.5% [9,32,71,73,74]. In this study, MC occurred in 3% of episodes, with *C. albicans* being the most frequently isolated in combination with other *Candida* spp., consistent with the rates and distribution reported in previous multicentre studies conducted worldwide [32,75,76,77,78]. Nevertheless, only 4/28 (14%) of the participating hospitals used chromogenic media, thus the incidence of MC might have been underestimated. Indeed, the proportion of MC cases in the present survey was slightly lower than a recent study carried out in a Greek tertiary care academic hospital during the same period, which detected a rate of 4.7% [9]. It is noteworthy that detection of MC remains crucial for optimum treatment, since species with intrinsic resistance or reduced susceptibility to different antifungals may be among the causative pathogens, as previously described [71]. In fact, 4/24 (17%) MCs in the present study were caused by a common (2 *C. albicans*, 2 *C. parapsilosis* SC) and a rare *Canidida* spp. (2 *C. kefyr*, 1 *C. lusitaniae*, 1 *C. sake*), limiting therapeutic options [54]; FLC-susceptible-dose dependent/resistant *C. parapsilosis* SC were isolated from 3/24 (12%) patients presented with *C. albicans*-*C. parapsilosis* SC coinfection; and 1/24 (4%) MC was caused by *C. albicans*-*C. krusei*, with the latter having an inherent instinct of resistance to FLC, indicating the importance of using chromogenic media to detect MCs. Hence, special attention is required for isolating mixed *Candida* spp., particularly in immunocompromised patients.

The overuse of ineffective or unnecessary antifungal therapy is a prime mover for the emergence of resistance in *Candida* spp. Of note, the expanding incidence of *Candida* BSIs due to azole- and echinocandin-resistant isolates is considered a serious public health threat [79]. Despite the concern for a genetic predisposition to nonsusceptibility to echinocandins, several surveys have revealed that azole resistance occurs more commonly in *C. parapsilosis* SC [80,81]. Indeed, while resistance to echinocandins among *C. parapsilosis* SC isolates was not observed, in line with recent global antifungal surveillance results [82], the rate of FLC resistance was 20%. A short time ago, FLC resistance was considered to be uncommon among *C. parapsilosis* SC strains. While the phenomenon had appeared to be restricted to certain geographic regions, an ever-growing number of FLC-resistant clinical isolates have been reported worldwide from Brazil [57,83,84], France [85], India [86], Korea [87,88], Kuwait [32], North America [82], and South Africa [56,89], whilst the problem is already broadly disseminated in Italy [48,82,90] and Turkey [91,92,93]. FLC resistance *C. parapsilosis* SC has been well associated with selective drug pressure attributable to the extensive use of FLC as prophylaxis and treatment or even to exposure to systemic antimicrobials [94,95] with subsequent patient-to-patient spread in an epidemic way with clonal transmission and establishment of persistent resistant isolates within the hospital environment [83,84,85,92]. In the present study, FLC-resistant *C. parapsilosis* SC strains were found in all medical units of hospitals geographically distributed all over the country. Of note, in light of concerns about decreased susceptibility of *C. parapsilosis* SC to echinocandins, both the American [6] and the European [96] guidelines have recommended the use of FLC coupled with prompt catheter removal in patients with *C. parapsilosis* SC BSIs. At the same time, FLC is still the most intensely prescribed antifungal agent in the Greek hospitals [97]; considering its lower price compared with echinocandins, usually its administration may be preferable for *C. parapsilosis* SC-driven candidemias under European guidelines [96], while the overuse of antimicrobials in Greece is threatening [97]. These factors may be associated with the high rate of FLC-resistant *C. parapsilosis* SC bloodstream isolates reported in the present study. Worryingly, a significant increase in their incidence rate was identified over time (from 0% in 2009 to 32% in 2018, *p* < 0.0001), as previously described [74,87], whereas FLC-VRC cross-resistance occurred in a high proportion of such isolates, being consistent with previous findings [54,74,85,92,98]. Notably, a trend of higher mortality rates in candidemic patients infected with FLC-resistant *C. parapsilosis* SC than in those with FLC-susceptible (42–50% versus 16–26%, respectively) has been described [87,92]. Therefore, it is highly important to diligently monitor the local burden of antifungal resistance, which can undermine the clinical efficacy of commonly used antifungals and may have detrimental consequences that can potentially result in longer hospital stay, higher costs, and poor outcome.

On the other hand, echinocandins are considered to be the first-line treatment of *C. glabrata* SC-driven BSIs, since this particular species generally presents reduced susceptibility to azoles. Nevertheless, *C. glabrata* SC rapidly acquires resistance to echinocandins after repeated or long-term exposure to this antifungal class [99,100]. The present data are comparable with several population-based studies demonstrating 3–5% echinocandin resistance among *C. glabrata* SC isolates [101]. The fact that all strains exhibiting cross-resistance to all three echinocandins, but not to azoles, were recovered from ICU patients, might be explained by the extensive use of echinocandins in the Greek ICU settings [62]. However, one should keep in mind that many centres worldwide have reported resistance rates of 10–15% [102,103], while echinocandin resistance in *C. glabrata* SC is related to clinical failure [103]. This underscores the need for implementation of local guidelines, as recently suggested [104], in the context that empirical antifungal strategies should be tailored to the nosocomial setting, highlighting once again the central role of epidemiological surveillance to avoid the development of resistant isolates.

Resistance to antifungal drugs poses a tremendous challenge in the treatment of invasive fungal infections. Despite a long history of clinical use, resistance to polyenes remains low [105]. Indeed, our data indicated a high rate (100%) of in vitro susceptibility to amphotericin B (AMB) across all *Candida* spp. As a class, polyenes have an extended antifungal spectrum that covers most clinically relevant yeasts and moulds. Although conventional AMB deoxycholate has been associated with substantial toxicities, its lipid derivatives, particularly liposomal AMB, have an improved safety profile. Liposomal AMB (3 mg/kg/day) has been shown to be as effective as micafungin for treatment of *Candida* BSIs [106] and its use should be considered when there is a history of intolerance to echinocandins and/or azoles, the infection is refractory to other therapy, or the causative isolate is resistant to other agents [6]. Thus, adapting the current guidelines to local ecology described by increased levels of FLC and/or echinocandin resistance in certain settings, liposomal AMB may play a key role in the empirical treatment of candidemia at local scale.

Limitations of the present study comprise potential differences in clinical and laboratory practices across the participating hospitals as well as the lack of detailed individual patient data (demographics, comorbidities, previous antifungal drug exposure, risk factors for candidemia, outcome) given its retrospective nature. Nevertheless, the study fills a gap in the existing literature and provides a large contemporary overview on *Candida* BSIs from an entire country over a 10-year period, which can be instrumental in designing local therapeutic strategies and provide a point of reference paving the way for subsequent epidemiological surveys. 

## 5. Conclusions

The present study provides a much-needed updated view of the epidemiology of candidemia in the Greek general patient population at national level since 2009. The incidence of candidemia increased significantly during the last decade and a species shift toward *C. parapsilosis* SC was observed. Although antifungal resistance levels remain relatively low in the total sample, the increase of FLC-resistant *C. parapsilosis* SC raises concern, pointing out the desirability of systematic susceptibility testing of all *Candida* bloodstream isolates so as to monitor and detect significant changes in trends. Overall, clinical recommendations must be balanced by epidemiological concerns. The present findings underscore the need for increased awareness, introduction of antifungal stewardship programmes, and strict implementation of infection control measures to diminish the incidence and resistance rates of *Candida* BSIs seen to be rising in Greece. 

## Figures and Tables

**Figure 1 jof-08-00116-f001:**
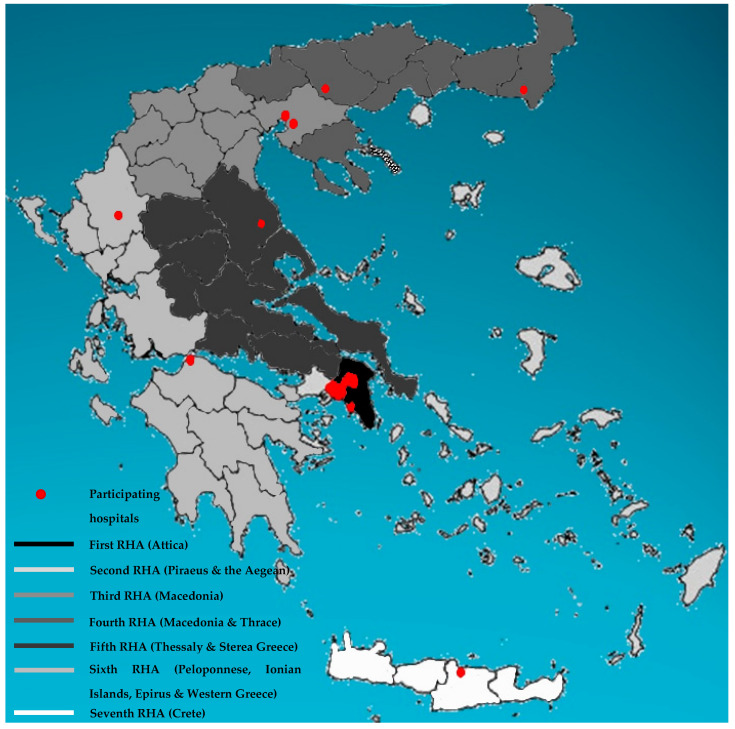
Geographic distribution of the 28 participating hospitals (red dots) across the seven Greek regional health authorities (RHA).

**Figure 2 jof-08-00116-f002:**
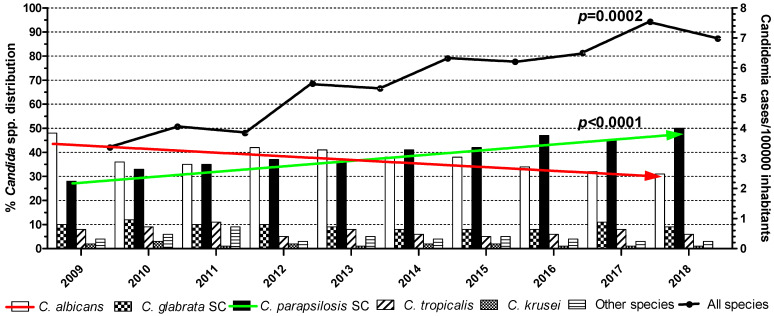
Species distribution of *Candida* bloodstream isolates and temporal changes in candidemic episodes per 100,000 inhabitants. Statistically significant differences between the isolation rates of *C. albicans* (red falling arrow) and *C. parapsilosis* species complex (SC; green rising arrow) (*p* < 0.0001) and in the incidence of candidemia (*p* = 0.0002) were recorded over the years.

**Figure 3 jof-08-00116-f003:**
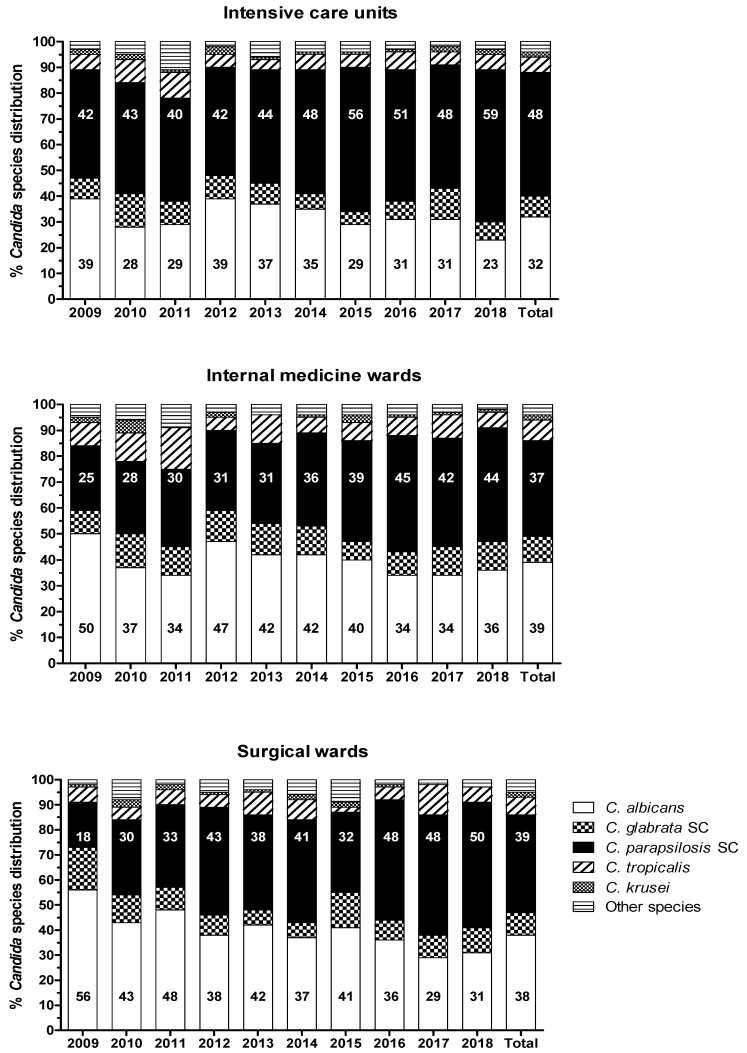
Trends in the ward-wise distribution of *Candida* spp. over the years. Statistically significant differences between *C. albicans* and non-*albicans Candida* spp. were recorded in all medical units (*p* < 0.0001), mainly due to the notable increases in the frequency of *C. parapsilosis* species complex (SC) isolation.

**Figure 4 jof-08-00116-f004:**
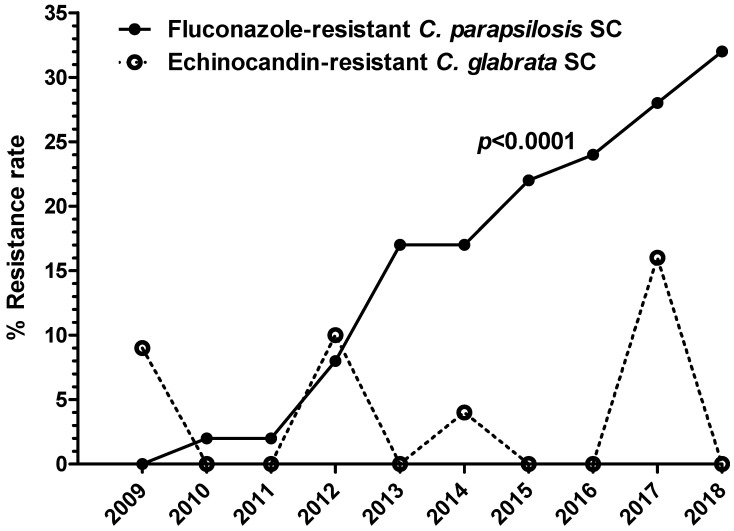
Trends of *C. parapsilosis* species complex (SC) to fluconazole resistance and of *C. glabrata* SC to echinocandin resistance. Statistically significant increase of fluconazole-resistant *C. parapsilosis* SC isolates was observed during the study period (*p* < 0.0001).

**Table 1 jof-08-00116-t001:** In vitro susceptibility profile of *Candida* bloodstream isolates collected during 2009–2018 to nine antifungals. Susceptibility testing was performed as per hospital protocol.

*Candida* spp. and Antifungal Agent	No of Isolates	Clinical Breakpoints *			ECVs/ECOFFs *	
		S	I/SDD	R	WT	Non-WT
		No (%)	No (%)	No (%)	No (%)	No (%)
*C. albicans*						
Anidulafungin	724	724 (100)	0 (0)	0 (0)	-	-
Caspofungin	1883	1794 (95)	24 (1)	50 (3)	15 (1)	-
Micafungin	1883	1846 (98)	25 (1)	12 (1)	-	-
Flucytosine	1868	-	-	-	1868 (100)	0 (0)
Fluconazole	1883	1793 (95)	41 (2)	49 (3)	-	-
Itraconazole	709	-	-	-	681 (96)	28 (4)
Posaconazole	724	15 (2)	-	-	690 (95)	19 (3)
Voriconazole	1883	1732 (92)	94 (5)	57 (3)	-	-
Amphotericin B	1883	-	-	-	1883 (100)	0 (0)
*C. parapsilosis* SC						
Anidulafungin	396	396 (100)	0 (0)	0 (0)	-	-
Caspofungin	2216	2216 (100)	0 (0)	0 (0)	-	-
Micafungin	2216	2216 (100)	0 (0)	0 (0)	-	-
Flucytosine	2189	-	-	-	2189 (100)	0 (0)
Fluconazole	2216	1717 (78)	58 (2)	441 (20)	-	-
Itraconazole	369	-	-	-	342 (93)	27 (7)
Posaconazole	369	-	-	-	376 (95)	20 (5)
Voriconazole	2216	2027 (92)	163 (7)	26 (1)	-	-
Amphotericin B	2216	-	-	-	2216 (100)	0 (0)
*C. glabrata* SC						
Anidulafungin	203	196 (97)	0 (0)	7 (3)	-	-
Caspofungin	500	486 (97)	5 (1)	9 (2)	-	-
Micafungin	500	485 (97)	3 (1)	12 (2)	-	-
Flucytosine	500	-	-	-	500 (100)	0 (0)
Fluconazole	500		477 (95)	23 (5)	-	-
Itraconazole	203	-	-	-	188 (93)	15 (7)
Posaconazole	203	-	-	-	179 (88)	24 (12)
Voriconazole	500	-	-	-	470 (94)	30 (6)
Amphotericin B	500	-	-	-	500 (100)	0 (0)
*C. tropicalis*						
Anidulafungin	75	75 (100)	0 (0)	0 (0)	-	-
Caspofungin	373	366 (98)	0 (0)	7 (2)	-	-
Micafungin	373	370 (99)	0 (0)	3 (1)	-	-
Flucytosine	373	-	-	-	373 (100)	0 (0)
Fluconazole	373	322 (86)	28 (8)	23 (6)	-	-
Itraconazole	75	-	-	-	75 (100)	0 (0)
Posaconazole	75	-	-	-	75 (100)	0 (0)
Voriconazole	373	330 (88)	38 (10)	5 (1)	-	-
Amphotericin B	373	-	-	-	373 (100)	0 (0)
*C. krusei*						
Anidulafungin	33	33(100)	0 (0)	0 (0)	-	-
Caspofungin	77	74 (96)	1 (1)	2 (3)	-	-
Micafungin	77	77 (100)	0 (0)	0 (0)	-	-
Flucytosine	77	-	-	-	77 (100)	0 (0)
Fluconazole	77	-	-	-	77 (100)	0 (0)
Itraconazole	33	-	-	-	33 (100)	0 (0)
Posaconazole	33	-	-	-	33 (100)	0 (0)
Voriconazole	77	77 (100)	0 (0)	0 (0)	-	-
Amphotericin B	77	-	-	-	77 (100)	0 (0)
Total						
Anidulafungin	1431	1424 (99.5)	0 (0)	7 (0.5)		
Caspofungin	5049	4928 (97.6)	38 (0.8)	68 (1.3)	15 (0.3)	
Micafungin	5049	4981 (98.7)	41 (0.8)	27 (0.5)		
Flucytosine	5007				5007 (100)	0 (0)
Fluconazole	5049	3832 (75.9)	604 (12)	536 (10.6)	77 (1.5)	
Itraconazole	1389				1319 (95)	70 (5)
Posaconazole	1431	15 (1)			1353 (94.6)	63 (4.4)
Voriconazole	5049	4166 (82.6)	295 (5.8)	88 (1.7)	470 (9.3)	30 (0.6)
Amphotericin B	5049				5049 (100)	0 (0)

**Abbreviations**: SC: species complex, ECVs/ECOFFs: epidemiological cut-off values, S: susceptible, I: intermediate, SDD: susceptible-dose dependent, R: resistant, WT: wild type. * CLSI [19]/EUCAST [13] clinical breakpoints and method-specific ECVs/ECOFFs (CLSI [20], EUCAST [13], Etest [21,22,23], Sensititre [22,24,25]) were used.

## Data Availability

Data available on request.

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
