# Peer review of "Increasing Incidence and Shifting Epidemiology of Candidemia in Greece: Results from the First Nationwide 10-Year Survey"

_jof, 2022, doi:10.3390/jof8020116_

Round 1

Reviewer 1 Report

The authors have reported the epidemiological data for candidemia from 28 Greek centers for the period of 2009-2018. The data are nationwide and provide a fair perspective for the temporal changes in terms of species distributions and antifungal resistance rates.

Some comments and inquiries:

1. C. parapsilosis SC has been detected as the most common SC causing candidemia. Are there any available data from any of the centers and over the years that are related to clonal relationship? This may be very helpful to suggest/exclude existence of any possible ongoing outbreak through the years.

2. Line 39. Invasive candidiasis is the most common invasive fungal infection observed worldwide. Except for the outbreaks, “most” of these Candida infections are endogenous, not hospital-acquired. Whether the Candida infection is potentially endogenous or exogenous also relies on the infecting species, as has also been addressed by the authors in the Discussion section. The related statement (line 39) appears to be incomplete at its present form and needs to be modified and detailed accordingly not to be misleading.

3. Lines 92-94. Relevant reference(s) should be cited for the definition of “ICU-acquired candidemia.”

4. Lines 112-124. Although some of the commercial systems may provide comparable MIC data with the reference methods in general, incompatible results may be observed for a number of strains. Further, MIC interpretive breakpoints and ECVs/ECOFFs are “method-specific”. For such nationwide studies aiming to determine resistance rates, all isolates should be tested with the very same in vitro antifungal susceptibility testing method; with one of the (CLSI or EUCAST) reference methods. Were the resistant/non-WT strains tested for confirmation with a reference method at the coordinating center?

5. Lines 105-111. Identification methods are very heterogeneous, as also addressed by the authors as a limitation (Discussion section).

6. Page 8. Echinocandins. EUCAST-AFST Subcommittee recommends not to test “caspofungin” due to reproducibility problem detected in multicenter studies. Instead, micafungin or anidulafungin are recommended to be tested to represent echinocandin class.

7. Page 14. Paragraph 1. Line 11. Reference no. 47 is a report from Italy.

Author Response

Responses to Reviewers’ Comments (Manuscript ID jof-1545849)

We would like to thank the Reviewers for their constructive comments. In the following, we address their concerns, point-by-point.

 Reviewer 1

  1. parapsilosis SC has been detected as the most common SC causing candidemia. Are there any available data from any of the centers and over the years that are related to clonal relationship? This may be very helpful to suggest/exclude existence of any possible ongoing outbreak through the years.

Response: We thank the reviewer for pointing this out. Indeed, clonal candidemia outbreaks by C. parapsilosis have been reported in diverse geographical regions, with most of them occurring in ICUs. In the present study, “the horizontal transmission potential of C. parapsilosis species complex via contaminated medical equipment and hands of health care personnel, which may lead to crossover infections between patients and is often responsible for nosocomial cluster outbreaks, cannot be excluded” (page 13, 1st paragraph, lines 10-13). Unfortunately, given the retrospective nature of the study and the fact that application of molecular techniques in Medical Mycology have not yet been implemented in most hospital-based Greek clinical laboratories (page 13, 2nd paragraph, lines 9-10), there are no available data related to clonal relationship. Nevertheless, “C. parapsilosis species complex strains were found in all medical units of hospitals geographically distributed all over the country” (page 13, 1st paragraph, lines 16-17) throughout the decade (Figure 3), indicating that the notable increases in the frequency of C. parapsilosis species complex isolation may not be hospital-specific.

Line 39. Invasive candidiasis is the most common invasive fungal infection observed worldwide. Except for the outbreaks, “most” of these Candida infections are endogenous, not hospital-acquired. Whether the Candida infection is potentially endogenous or exogenous also relies on the infecting species, as has also been addressed by the authors in the Discussion section. The related statement (line 39) appears to be incomplete at its present form and needs to be modified and detailed accordingly not to be misleading.

Response: The reviewer is right. The sentence is now changed to “Candidemia is among the leading nosocomial bloodstream infections (BSIs) globally, representing the most frequently encountered manifestation of invasive candidiasis.”

Lines 92-94. Relevant reference(s) should be cited for the definition of “ICU-acquired candidemia.”

Response: We appreciate for the suggestion given by the reviewer. A reference (Bassetti M. et al. Incidence and outcome of invasive candidiasis in intensive care units (ICUs) in Europe: results of the EUCANDICU project. Crit Care. 2019) is now included.

 Lines 112-124. Although some of the commercial systems may provide comparable MIC data with the reference methods in general, incompatible results may be observed for a number of strains. Further, MIC interpretive breakpoints and ECVs/ECOFFs are “method-specific”. For such nationwide studies aiming to determine resistance rates, all isolates should be tested with the very same in vitro antifungal susceptibility testing method; with one of the (CLSI or EUCAST) reference methods. Were the resistant/non-WT strains tested for confirmation with a reference method at the coordinating center?

Response: We fully agree with the reviewer’s concern. Unfortunately, due to the retrospective nature of our study and mainly the lack of routine storage of clinical isolates by the majority of the participating hospitals (>90%), verification of resistant/non-WT phenotypes using a reference method (CLSI/EUCAST) was not feasible. Of note, the heterogeneity in laboratory practices is already addressed in the discussion section as a potential limitation. 

Lines 105-111. Identification methods are very heterogeneous, as also addressed by the authors as a limitation (Discussion section).

Response: We acknowledge and agree with the remark of the reviewer. In accordance with the aforementioned, confirmation using a single identification method at a coordinating center was not possible.

Page 8. Echinocandins. EUCAST-AFST Subcommittee recommends not to test “caspofungin” due to reproducibility problem detected in multicenter studies. Instead, micafungin or anidulafungin are recommended to be tested to represent echinocandin class.

Response: The reviewer is right. The word “cross-resistance” is now changed to “elevated MIC values”.

Page 14. Paragraph 1. Line 11. Reference no. 47 is a report from Italy.

Response: This is now corrected.

Reviewer 2 Report

Authors need to correct some technical errors.

Author Response

We would like to thank the Reviewers for their constructive comments. In the following, we address their concerns, point-by-point.

Reviewer 2

Line 15: Globally, Candidemia displays…

Response: This is now corrected.

Line 55: Although Candida albicans…

Response: This is now corrected.

Reviewer 3 Report

In this work, it is described the current tendency of infections caused by Candida in Greece. The article is well written and the results are clearly presented. However, there are two points that I’d like to highlight:

  • Figure 2: I did not understand what the two arrows for albicans and C. parapsilosis. What do the authors mean? It should represent that the number of infections caused by C. parapsilosis is increasing while C. albicans is decreasing. It is not clear and I am not sure if those arrows are needed if that is what the authors want to say
  • Figure 4: why are only shown the results for parapsilosis and C. glabrata? Please specify. Also did the authors find an explication for the strange up-and-down tendency of resistance to equinocandins?

Author Response

We would like to thank the Reviewers for their constructive comments. In the following, we address their concerns, point-by-point

Reviewer 3

In this work, it is described the current tendency of infections caused by Candida in Greece. The article is well written and the results are clearly presented.

Response: We are grateful for the reviewer's supportive comment on our manuscript.

 Figure 2: I did not understand what the two arrows for albicans and C. parapsilosis. What do the authors mean? It should represent that the number of infections caused by C. parapsilosis is increasing while C. albicans is decreasing. It is not clear and I am not sure if those arrows are needed if that is what the authors want to say.

Response: We acknowledge the remark of the reviewer. Indeed, the arrows have been added to highlight the statistically significant reduction in C. albicans rates in parallel with a double increase in C. parapsilosis SC rates over the decade, as indicated by the corresponding p value (<0.0001). This is now clarified in the figure legend.

Figure 4: why are only shown the results for parapsilosis and C. glabrata? Please specify. Also did the authors find an explication for the strange up-and-down tendency of resistance to equinocandins?

Response: As stated in the last paragraph of the results section (page 8) “No significant trend over time was found in the annual susceptibility rates for the rest of the antifungals and Candida spp.”, and thus are not presented. The up-and-down tendency of echinocandin-resistant C. glabrata species complex isolates may be explained by the total low number of isolates for which susceptibility data were available. Namely, the 9%, 10%, 4% and 16% echinocandin resistance observed during 2009, 2012, 2014 and 2017 correspond to 1/11, 2/22, 1/25 and 3/19, respectively, C. glabrata species complex strains. Of note, “… isolates were distributed equally through the years of the study period (0-3 isolates annually; p=0.80)”, as mentioned in page 8 (4th paragraph, lines 8-9)

Reviewer 4 Report

In the paper the authors speak about an increased incidence 
 of candidemia in Greece with a species shift toward C. parapsilosis.

The paper is well written but I have some comments:

Introduction: 

  • line 57 add a short paragraph to interline the importance of C. auris infection and candidemia. Add the following reference: 10.3390/jof7100878 and 10.3390/antibiotics9110778.
  • Add a paragraph to interline the importance of Biofilm add the reference Cala et al Candida parapsilosis Infection: A Multilocus Microsatellite Genotyping-Based Survey Demonstrating an Outbreak in Hospitalized Patients. Annals of clinical and laboratory science Volume 50, Issue 5, Pages 657 - 6641 September 2020
  • Add the conclusion section.

Author Response

We would like to thank the Reviewers for their constructive comments. In the following, we address their concerns, point-by-point.

Reviewer 4

In the paper the authors speak about an increased incidence of candidemia in Greece with a species shift toward C. parapsilosis. The paper is well written.

Response: We are grateful for the positive remarks regarding our manuscript.

Line 57: add a short paragraph to interline the importance of C. auris infection and candidemia. Add the following reference: 10.3390/jof7100878 and 10.3390/antibiotics9110778.

Response: We acknowledge the remark of the reviewer. As already stated, “… novel pathogenic species with multi-resistance profiles, such as C. auris, set a worrisome trend and amplify the call for alertness (Rhodes J. and Fisher M.C. Global epidemiology of emerging C. auris. Curr. Opin. Microbiol. 2019)” (lines 59-60). In fact, the introduction section clarifies the motivation for the work presented and serves as the roadmap for the manuscript by clearly stating the study's background and aims. Nevertheless, neither the epidemiology, management strategies and patient outcomes of invasive C. auris infections (10.3390/jof7100878) nor the available methods for screening, detection and testing of C. auris isolates (10.3390/antibiotics9110778) is the subject matter of the present study.  

Add a paragraph to interline the importance of Biofilm add the reference Cala et al Candida parapsilosis Infection: A Multilocus Microsatellite Genotyping-Based Survey Demonstrating an Outbreak in Hospitalized Patients. Annals of clinical and laboratory science Volume 50, Issue 5, Pages 657 - 6641 September 2020

Response: As stated in page 12 (last paragraph, lines 1-3) “The dominance of C. parapsilosis SC is worrisome since it has been associated with central venous catheter infections and the administration of parenteral nutrition due to its propensity to form tenacious biofilms”. Indeed, several studies have shown that Candida spp. can easily adhere to and form biofilms on inert surfaces, while clinical C. parapsilosis isolates feature distinct biofilm structures. However, the importance and clinical aspects of Candida biofilms is beyond the scope of our study. On the other hand, the phenomenon of clonal candidemia outbreaks by C. parapsilosis has already been reported in the discussion section accompanied by corresponding references. In particular, “the horizontal transmission potential of C. parapsilosis species complex via contaminated medical equipment and hands of health care personnel, which may lead to crossover infections between patients and is often responsible for nosocomial cluster outbreaks” (page 13, 1st paragraph, lines 10-13) and “... subsequent patient-to-patient spread on an epidemic way with clonal transmission and establishment of persistent resistant isolates within the hospital environment [83–85,92]” (page 14, 1st paragraph, lines 14-16).

Add the conclusion section.

Response: Based on the manuscript preparation guidelines, the conclusion section is not mandatory. Nevertheless, given the length of the discussion section, we appreciate for the suggestion given by the reviewer and we fully agree that a separate section placed at the end of the manuscript will help the reader not to lose sight of the main messages of the study. Thus, the heading “Conclusions” is now added in the concluding paragraph of the discussion section, which provides a summarization along with the significance, implications and potential next steps.

Round 2

Reviewer 1 Report

All of my critiques have been addressed